

# MSR-UNet: enhancing multi-scale and long-range dependencies in medical image segmentation

Shuai Wang[1], Lei Liu[1,2], Jun Wang[3], Xinyue Peng[1] and Baosen Liu[4]

[1] School of Computer Science and Technology, Huaibei Normal University, Huaibei, China
[2] Huaibei Key Laboratory of Digital Multimedia Intelligent Information Processing, Huaibei, China
[3] College of Electronic and Information Engineering, Hebei University, Baoding, China
[4] Huaibei People's Hospital, Huaibei, China

Corresponding author
Lei Liu, liul@chnu.edu.cn

## ABSTRACT

Transformer-based technology has attracted widespread attention in medical image segmentation. Due to the diversity of organs, effective modelling of multi-scale information and establishing long-range dependencies between pixels are crucial for successful medical image segmentation. However, most studies rely on a fixed single-scale window for modeling, which ignores the potential impact of window size on performance. This limitation can hinder window-based models' ability to fully explore multi-scale and long-range relationships within medical images. To address this issue, we propose a multi-scale reconfiguration self-attention (MSR-SA) module that accurately models multi-scale information and long-range dependencies in medical images. The MSR-SA module first divides the attention heads into multiple groups, each assigned an ascending dilation rate. These groups are then uniformly split into several non-overlapping local windows. Using dilated sampling, we gather the same number of keys to obtain both long-range and multi-scale information. Finally, dynamic information fusion is achieved by integrating features from the sampling points at corresponding positions across different windows. Based on the MSR-SA module, we propose a multi-scale reconfiguration U-Net (MSR-UNet) framework for medical image segmentation. Experiments on the Synapse and automated cardiac diagnosis challenge (ACDC) datasets show that MSR-UNet can achieve satisfactory segmentation results. The code is available at https://github.com/davidsmithwj/MSR-UNet (DOI: 10.5281/zenodo.13969855).

# INTRODUCTION

Medical image segmentation, a crucial task in computer vision and medical image analysis, aims to accurately identify regions of clinical significance within medical images. Its main goal is to provide an authentic foundation for clinical diagnosis and pathological studies, thereby assisting doctors and medical experts in making accurate judgments. Medical image segmentation and analysis are widely used in various medical fields, including computer-aided surgery (*Félix et al., 2020*; *Huo et al., 2019*; *Elbatel, Martí & Li, 2024*),

pathological analysis (*Chen et al., 2020*), and clinical diagnosis (*Azad et al., 2022*), offering valuable support for medical practice (*Feng et al., 2020*; *Gröhl et al., 2024*; *Hou et al., 2023*).

Over the past two decades, the rapid advancement of deep learning has led to the emergence of neural network-based methods as the dominant technology in medical image processing. Most of these methods are built on convolutional neural networks (CNNs), especially U-Net (*Ronneberger, Fischer & Brox, 2015*) and its various derivatives (*Jiang, Chen & Tian, 2023*; *Wang et al., 2024*; *Armand et al., 2024*), which have made significant strides in recent years. The success of these methods stems from their encoder-decoder structure, where skip connections effectively integrate multi-scale features from the encoder with semantic features from the decoder. However, convolution operations are inherently limited by their small-scale receptive fields and cannot establish long-range dependencies between pixels—an essential requirement in certain segmentation tasks (*Hou et al., 2020*; *Huang et al., 2019*). Because of the variability in organ structures and pathological conditions, CNN-based methods often struggle with complex medical image segmentation challenges, such as significant variations in size, shape, and texture across different individuals (*Lin et al., 2022*).

Recently, vision transformers (ViTs) have gained widespread attention as alternatives to CNNs in visual recognition tasks, including medical image segmentation (*Dosovitskiy et al., 2020*; *Liu et al., 2021*; *Li et al., 2024*). ViTs are particularly well-suited for segmentation tasks due to their ability to capture long-range dependencies. For example, TransBTS (*Wang et al., 2021*) effectively combines 3D CNNs with transformers to solve the problems of multimodal brain tumor segmentation. Subsequently, various transformer-based architectures, such as PMTrans (*Zhang & Zhang, 2021*), TransUNet (*Chen et al., 2021*), UNet Transformer (UNETR) (*Hatamizadeh et al., 2022*), and MTANet (*Ling et al., 2024*), have been proposed for medical image segmentation. These transformer-based frameworks have outperformed traditional CNN-based methods, achieving state-of-the-art results across multiple medical image benchmarks.

Despite their success, a fundamental difference between transformers and CNN is in how they model dependencies. Swin-transformer-based medical image segmentation methods (*Lin et al., 2022*; *Cao et al., 2022*; *Hatamizadeh et al., 2021*), for example, emphasize capturing relationships within local regions, thus focusing on local contextual information. While this localized approach is effective in handling local structures, it may overlook long-range dependencies, which are important for accurately representing variations in organ size and shape. Neglecting these long-range correlations can reduce the model's global perception, potentially lowering its accuracy. Medical images often contain structures and features at various scales, and focusing exclusively on local regions may limit the ability to fully capture multi-scale information. This limitation can hinder the accurate segmentation of organs, lesions, or anatomical structures, as these tasks require a model to understand and utilize both local and global context to precisely segment the target regions.

Therefore, it is essential to investigate how to balance the modeling of multi-scale information and long-range dependencies. In this article, we propose a novel self-attention mechanism called multi-scale reconfiguration self-attention (MSR-SA). The core

 

operations of MSR-SA focus on the reconfiguration process, aiming to meet the specific requirements for modeling multi-scale and long-range relationships, even at the expense of local window size. The traditional attention matrix is first decomposed into several sparse sub-attention matrices, and the original attention matrix is effectively approximated by superimposing these submatrices. Specifically, for a given input medical image, each discrete point is treated as a query. To gather keys, the input image is divided into several non-overlapping windows. Dilated sampling is then performed on each window, uniformly sampling a fixed number of points, followed by feature fusion across sampling points at the same position from different windows. Although the number of keys is limited, each key contains information across multiple windows of the entire image, enabling the acquisition of long-range dependencies within the constraints of local windows. To address the potential loss of local detail, which may occur when fusing information from multiple points, we incorporated dilated convolution into the multihead attention mechanism. Specifically, the size of the local windows increases gradually for different heads, and the dilation rate of point sampling is adjusted accordingly to maintain a consistent number of keys across all heads. Therefore, the acquired keys contain features from fewer locations, providing better local detail without introducing extra computational complexity. By integrating the features from multiple heads, MSR-SA effectively captures both multi-scale information and long-range dependencies simultaneously.

## Our contributions

Connecting the proposed MSR-SA attention mechanism to the MISSFormer backbone network (*Huang et al., 2023*) results in the development of our multi-scale reconfiguration U-Net, termed MSR-UNet. In summary, our main contributions are as follows:

- We propose a novel MSR-SA attention mechanism that groups ordinary attention heads, allocates increasing dilation rates, and divides each group into non-overlapping windows to aggregate multi-scale information. This mechanism simultaneously captures multi-scale information and long-range dependencies at a low cost in local windows.
- We introduce a new transformer-based segmentation framework, MSR-UNet, which extends the MSR-SA attention mechanism to develop a context bridge module using reconfiguration transformer blocks. This enables more effective capture of both local and global correlations within hierarchical multi-scale features.
- Benefiting from the two preceding innovations, MSR-UNet achieves promising segmentation performance on the widely used Synapse multi-organ dataset and the automated cardiac diagnosis challenge (ACDC) dataset, demonstrating the robustness and generalization of the proposed method.

The remainder of this article is organized as follows. The Related Works section introduces recent literature on deep learning based image segmentation, and the multi-scale feature integration and attention mechanism in transformer-bansed methods. The section of Methods provides a detailed description of the proposed MSR-UNet. The Experimental Results section demonstrate the implementation settings, applied datasets,

exhaustive segmentation results with analysis, and ablation studies. Finally, we summarize our work and reveal the directions of future research in the Conclusions section.

## RELATED WORKS

### Medical image segmentation

Early approaches mainly relied on CNNs, especially U-Net and its variants (*Alom et al., 2018*; *Zhou et al., 2020*; *Milletari, Navab & Ahmadi, 2016*), which adopt an encoder-decoder structure and have demonstrated excellent performance. UNet++ (*Zhou et al., 2018*) improved upon U-Net by introducing dense connections that allow the model to capture and extract feature information across various levels and scales. Attention U-Net (*Oktay et al., 2018*) incorporated an attention mechanism into U-Net, using an attention module to generate a gating signal that controls the importance of features at different spatial positions, while features from each resolution of the encoder are concatenated with the corresponding decoder features. Similarly, Res-UNet (*Xiao et al., 2018*) builds on U-Net by adding a weighted attention mechanism that enhances the learning of discriminative features between vascular and nonvascular pixels, thus better maintaining retinal vascular structures. R2U-Net (*Alom et al., 2018*) combines the strengths of U-Net, ResNet, and region-based CNN (RCNN), improving segmentation performance by refining the structure of U-Net without increasing the parameter count. Ki-Net (*Freudenthaler et al., 2022*) leverages the advantages of both Ki-Net for low-level fine edge feature mapping and U-Net for high-level shape feature mapping, leading to improved segmentation accuracy and fast convergence for small anatomical annotations and fuzzy noise boundaries. For 3D medical image segmentation, V-Net (*Milletari, Navab & Ahmadi, 2016*) proposed a volumetric and fully convolutional neural network, which was trained on MRI volumes to predict the whole volume segmentation. Based on the relatively fixed spatial relationship between internal structures in medical imnages, DARR (*Fu et al., 2020*) proposed an unsupervised domain adaptation method for 3D multi-organ segmentation. A jigsaw puzzle task and a super-resolution network were jointly trained with the segmentation task to guarantee the transfer ability of the spatial relationship among multiple domains.

In addition to CNN-based models, transformer-based frameworks have become increasingly popular (*Chen et al., 2021*; *Cao et al., 2022*). One notable example is TransUNet, proposed by *Chen et al. (2021)*, which adopts a hybrid CNN-transformer structure. This design leverages high-resolution spatial features from CNNs alongside global contextual information from transformers. Inspired by U-Net, TransUNet upsamples the self-attention features from the transformer and combines them with high-resolution CNN features, which are passed through skip connections in the encoder path, ensuring precise localization. This hybrid approach allows the model to benefit from both transformer capabilities and the strengths of CNNs in medical image segmentation tasks. Similarly, *Cao et al. (2022)* introduced Swin-UNet, based on the Swin-transformer (*Liu et al., 2021*), which reduces computational complexity by computing self-attention within local windows. *Huang et al. (2023)* advanced the field with the MISSFormer framework, which redefines the transformer block within the U-shaped encoder-decoder structure,

particularly through the feedforward network MixFFN. By reintegrating local and global contexts, MISSFormer effectively captures global dependencies while maintaining local detail, leading to enhanced segmentation performance.

## Multi-scale feature integration

Many ViT-based medical image segmentation methods struggle with processing multi-scale information, mainly due to their division of input images into fixed-size small blocks, which leads to the loss of valuable information (*Shamshad et al., 2023*). To overcome this shortcoming, *Ghiasi & Fowlkes (2016)* proposed a pyramid network architecture that combines multi-scale attention with CNN feature extraction, allowing multi-scale correlations to be captured through multiresolution image processing. *Ji et al. (2021)* introduced the multicompound transformer (MCTrans), which represents multi-scale convolutional features as feature sequences and performs attention calculations both within and across scales. Unlike previous methods, MCTrans accounts for multi-scale attention and enriches features by incorporating trainable proxy embeddings in self-attention and cross-attention mechanisms. *Xie et al. (2021)* proposed the CoTr method, which bridges CNNs and Transformers using a flexible self-attention mechanism that reduces model complexity while effectively utilizing multidimensional features for multi-scale feature computation. For semantic segmentation, several studies have developed high-level modules to better capture multi-scale features (*Ngo, Cha & Han, 2019*; *Liu et al., 2019*; *Zhang et al., 2020*).

## Attention mechanism

Attention mechanisms are crucial for image segmentation as they enhance the perception capabilities of networks across various CNN architectures by introducing self-attention mechanisms (*Wang et al., 2018*; *Vaswani et al., 2017*). However, traditional dense self-attention mechanisms often result in high computational costs and a large number of parameters, making them less practical for real-world medical applications. To overcome this challenge, advanced solutions have been proposed (*Wang et al., 2020*; *Tang et al., 2022*). Axial attention reduces complexity by sequentially computing attention along the horizontal and vertical axes within local windows, capturing the global context more efficiently (*Wang et al., 2020*). CCNet introduces cross-channel and bidirectional attention mechanisms to enhance the representational power of semantic segmentation networks (*Huang et al., 2019*). To further reduce computational complexity, MCANet (*Li et al., 2022*) proposes multimodal cross-attention, which effectively exploits global information while increasing segmentation accuracy and minimizing the number of parameters in dense self-attention mechanisms, making it suitable for practical medical applications.

Currently, while CNN-based methods have demonstrated impressive performance in the field of medical image segmentation, they still fall short of fully meeting the stringent segmentation accuracy requirements in practical medical applications. In medical image analysis, image segmentation remains a challenging task. Due to the inherent locality of convolution operations, CNN-based methods struggle to capture global semantic information and semantic information related to pixels that are relatively distant.

Additionally, convolution operations find it difficult to extract comparative boundary features. On the other hand, the attention mechanism has gained widespread application. As the core building block of visual transformers, attention serves as a powerful tool for capturing long-term dependencies. However, ordinary attention is a full attention mechanism that computes pairwise label affinities across all spatial positions, thus resulting in high computational complexity and significant memory usage (*Zhu et al., 2023*). It is worth mentioning that Swin-UNet (*Cao et al., 2022*) employs a pure vision transformer structure for feature extraction. Based on the U-Net network structure, Swin-UNet has achieved excellent performance in medical image segmentation, but the segmentation results of Swin-UNet network are rough at the edges compared to those of CNN-based methods. In comparison, our approach achieves good results in segmenting edges.

## METHODS

Figure 1 shows the framework of the proposed MSR-UNet, which follows an encoder-decoder architecture similar to that of many previous studies. In this section, we will first introduce the specific structure of the MSR-UNet framework. Following this, we will provide a detailed explanation of the core module in MSR-UNet—the multi-scale reconfiguration self-attention (MSR-SA).

### Framework of MSR-UNet

As shown in Fig. 1, the proposed MSR-UNet adopts a layered encoder-decoder design, incorporating a reconfiguration transformer context-bridging module (*Huang et al., 2023*). The input image is first divided into a series of small, overlapping blocks of size $4 \times 4$ to maintain the local continuity of the convolutional layers. These overlapping blocks are then fed into the encoder, where they are processed using the proposed MSR-SA module within the reconfiguration transformer block to generate multi-scale features. The encoder contains both reconfiguration transformer blocks and overlapping patch-merging layers. The MSR-SA module efficiently captures local context information and long-range dependencies with moderate computational cost, while the patch-merging layer generates downsampled features. This ensures effective feature extraction and information flow, thereby enhancing the model's performance.

The generated multi-scale features are then fed into the reconfiguration transformer context-bridging module to exploit the local and global relationships among the multiple scales. Specifically, these multi-scale features are flattened in the spatial dimension and projected onto a 2D feature map to ensure consistency with the channel dimension. The processed features are then passed to the decoder, which also incorporates reconfiguration transformer blocks. Finally, through patch expansion and multi-scale feature concatenation, the decoder outputs multi-scale, layered, discriminative segmentation features. This integration of information from different scales results in a more refined and expressive feature representation for the model.

During segmentation prediction, MSR-UNet employs multi-scale features and skip connections as inputs to the decoder. The decoder also contains reconfiguration

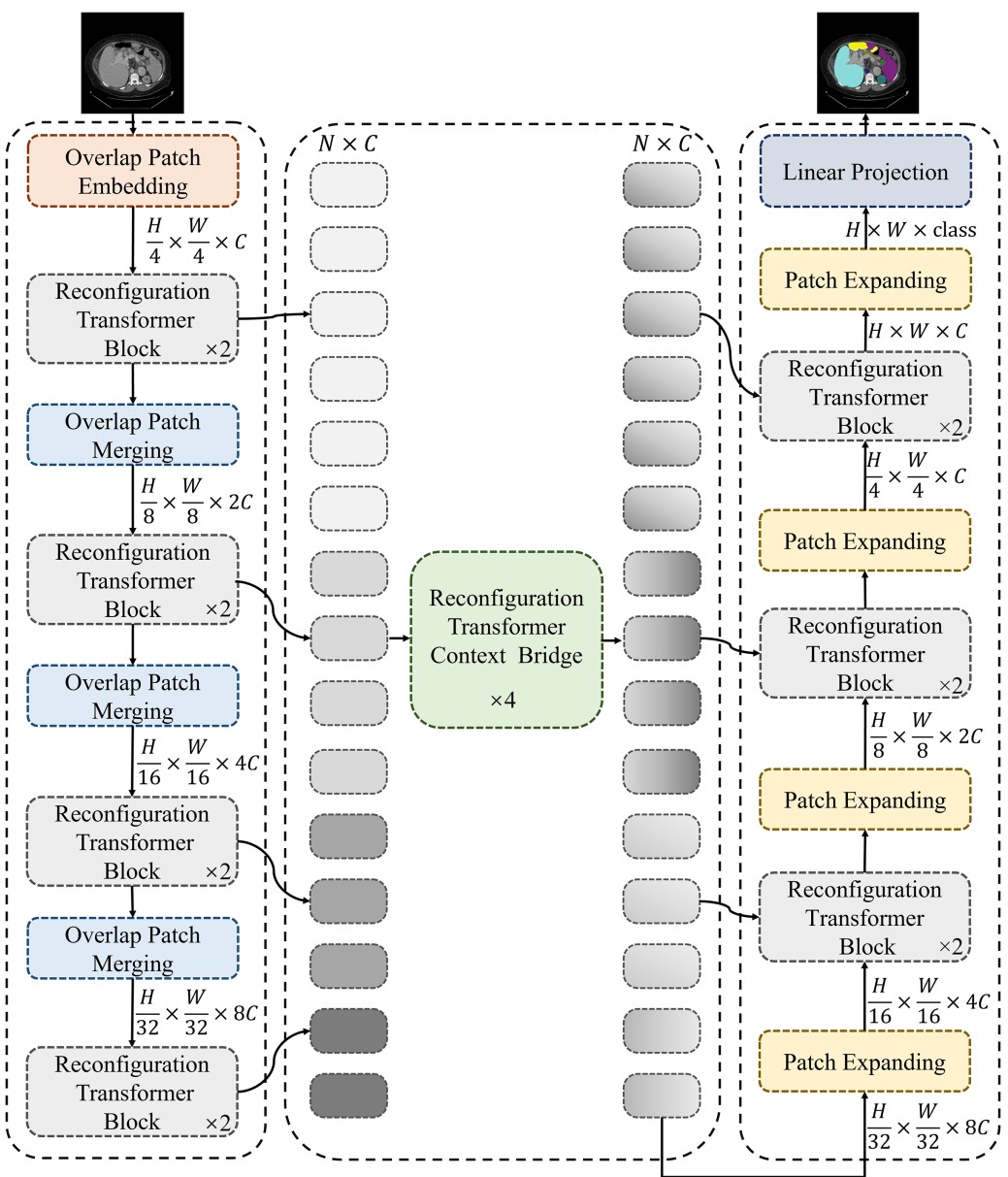

**Figure 1 Framework of the proposed MSR-UNet.** MSR-UNet consists of three parts: encoder, decoder, and context bridge module. These modules are all constructed based on reconfiguration transformer blocks.

transformer blocks and patch-expansion layers. Unlike the patch-merging layers, the patch-expansion layers upsample the adjacent feature map to twice its original resolution, ultimately producing a feature map that is four times the original resolution. This feature map is then projected back to the pixel level *via* linear projection to generate the final segmented image. By effectively combining features across different scales, this approach provides robust support for accurate pixel-level segmentation.

## MSR-SA module and reconfiguration transformer

Figure 2 depicts the details of the proposed MSR-SA module. The core idea behind MSR-SA is to integrate features of different granularities by decomposing a traditional attention matrix into a series of sparse sub-attention matrices. This decomposition allows for an approximate feature representation similar to the original attention matrix, while maintaining linear complexity relative to both computational cost and spatial resolution of the image being segmented. Specifically, the input feature map is evenly divided into multiple groups ($X \in R^{N \times C}$), and the self-attention of each group is computed separately to capture richer attention. The self-attention calculation consists of three steps: localization (dividing the entire feature map into several windows), dilated sampling (uniformly sampling a fixed number of points within each local window), and cross-window fusion (fusing the features of sampling points at the same positions from different windows). By fusing key features across multiple windows of the entire feature map, MSR-SA encapsulates long-range information. To address the potential loss of important features due to matrix sparsity in the generated key features, a dilated convolution is incorporated into the multihead attention mechanism. This strategy gradually increases the local window size and the dilation rate of point sampling for different groups. Despite changes in window size, this approach ensures a consistent number of sampling points and provides key features with multi-scale information. Below, we outline the implementation steps of MSR-SA.

First, the input feature map $X'$ is evenly divided into several groups along the channel dimensions as follows:

$$X' = \{X'_i \in R^{N' \times C'}, i = 1, ..., G\}, \tag{1}$$

where $X'_i$ is the feature map of the $i$-th group, $C' = C/G$, and $G$ is the number of groups. The divided features are assigned to different attention groups, with each group independently undergoing a reconfiguration process.

Second, the query (Q), key (K), and value (V) are generated separately for each attention group. For the $i$-th head attention, each point in $X'$ is treated as a query, and the query feature is obtained by applying a linear transformation as follows:

$$Q_i = W_i^Q X'_i \in R^{N' \times C'}, \tag{2}$$

where $W_i^Q \in R^{C' \times C'}$ represents a learnable linear embedding derived from a $1 \times 1$ convolution.

To balance the trade-off between modeling long-range dependencies and preserving the computational efficiency of local windows, the keys are collected in the following three steps:

(I) First, we apply a sliding window strategy to evenly divide $X'_i$ into multiple non-overlapping local windows. For simplicity, we assume that each local window has equal length and width, represented as $W = H$, and denoted by $S_i$. Thus, $X'_i$ can be expressed as follows:

$$X'_i = \{X^j_i \in R^{S_i \times S_i \times C'}, j = 1, ..., M_i\}, \tag{3}$$

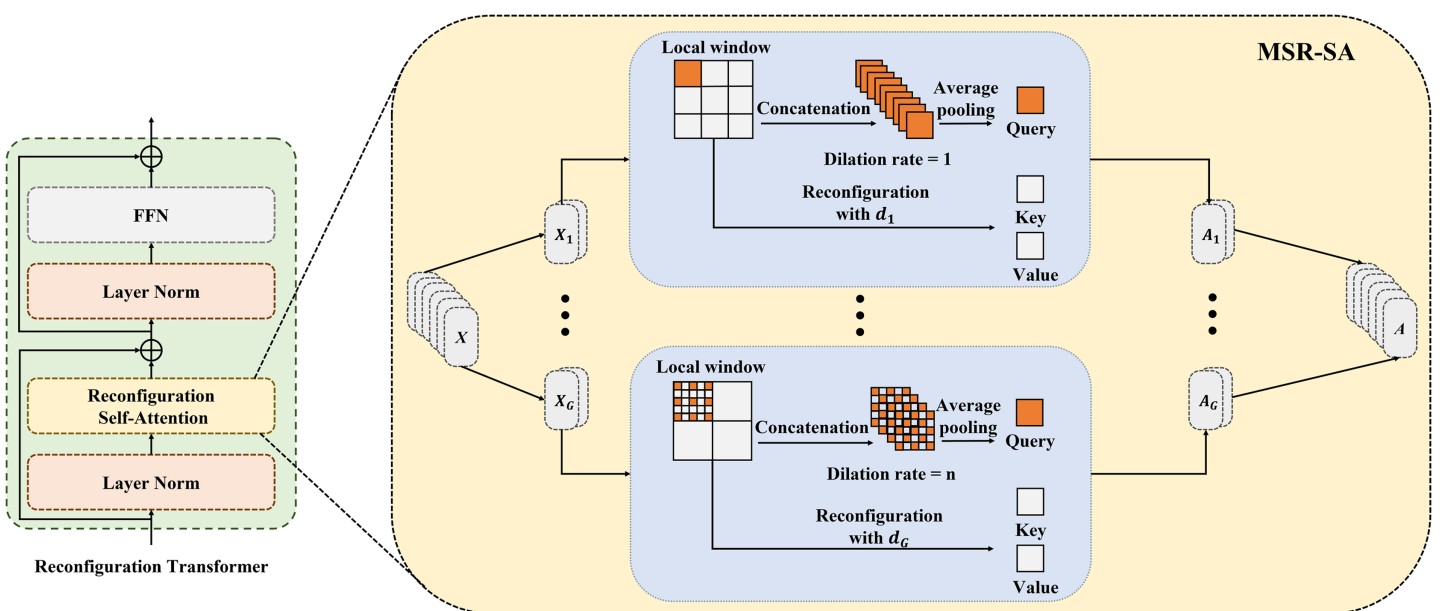

**Figure 2 Details of the proposed MSR-SA module.** Attention heads are divided into multiple groups and allocated increasing dilation rates. Each group is evenly partitioned into non-overlapping local windows and gathers same number of keys through dilated sampling, thereby obtaining both multi-scale and long-range information.

where $X_i^j$ indicates the feature map of the $j$-th window, and $M_i = H/S_i$ is the number of local windows in the $i$-th attention group. Notably, the size of the local windows is gradually increased in each group, resulting in a corresponding decrease in $M_i$.

(II) Next, a set of $M \times M$ points is uniformly sampled from each local window. In our experiments, $M$ was set to seven by default. For the $i$-th group, the set of sampling points $P_i$ is expressed as follows:

$$P_i = \{P_i^j \in R^{M \times M \times C'}, j = 1, \ldots, M_i\}, \tag{4}$$

where $P_i^j$ represents the sampling points in the $j$-th local window of the $i$-th group. Since different groups have varying local window sizes, it is crucial to ensure that the same number of sampling points is maintained across all local windows within each group. In our experiments, dilated sampling was adopted to adjust the dilation rates for different groups. The dilation rate corresponding to the $i$-th group is computed as follows:

$$D_i = \frac{S_i - 1}{M - 1}. \tag{5}$$

Therefore, the sampling points are uniformly distributed across all local windows. As the group index $i$ increases, the interval between sampling points widens, resulting in coarser information granularity.

(III) Previous studies have shown that restricting self-attention entirely within a local window can negatively impact modeling capabilities, mainly due to the lack of information interaction across local windows (*Liu et al., 2021*). To mitigate this issue and reduce the

number of key features, we introduced a cross-window fusion strategy. In this approach, feature embedding is performed at each sampling point:

$$K'_i = W_i^K P_i \in R^{M_i \times M^2 \times C'}, \tag{6}$$

$$V'_i = W_i^V P_i \in R^{M_i \times M^2 \times C'}, \tag{7}$$

where $K'_i$ and $V'_i$ represent the key and value of the $i$-th group, respectively. $W_i^K$ and $W_i^V$ represent the learnable linear embeddings in the $i$-th group, obtained from two independent $1 \times 1$ convolutions. The features of the sampling points at the same positions across different windows are then fused to acquire the final key and value features:

$$K_i = \sigma(K'_i) \in R^{M^2 \times C'}, \tag{8}$$

$$V_i = \sigma(V'_i) \in R^{M^2 \times C'}, \tag{9}$$

where $\sigma(\cdot)$ denotes the symmetric aggregation function, which achieves optimal results through maximum pooling.

In the operations described above, the original attention matrix is decomposed into a series of sparse sub-attention matrices. The fused features benefit from substantial long-range information, obtained by merging the features of the $M_i$ points that are uniformly distributed across the feature map. In addition, as the group index $i$ increases and the number of sampling points $M_i$ decreases, the fused features capture multi-scale details from fewer locations. The results indicate that these sub-attention matrices can effectively approximate the feature representation performance of an ordinary attention matrix.

Next, by performing self-attention on the collected query and key features for each group, we obtain:

$$X'_i = softmax\left(\frac{Q_i K_i^T}{\sqrt{h_i}}\right) V_i \in R^{H \times W \times C'}, \tag{10}$$

where $\sqrt{h_i}$ is the scaling factor. After performing self-attention on each group, we merge all the reconfigured attention groups to acquire the final output.

$$X' = \delta(X'_i, i = 1, \ldots, G) \in R^{H \times W \times C}, \tag{11}$$

where $\delta(\cdot)$ denotes the concatenation operation along the feature channel dimensions. Therefore, MSR-SA effectively captures long-range dependencies while simultaneously incorporating multi-scale information.

## Reconfiguration transformer context bridge block

In this section, the reconfiguration of the transformer block is integrated into the bridge module. As shown in Fig. 3, multilevel feature maps are obtained after feeding the patches through the encoder. For these multilevel feature maps, we first flatten their spatial dimensions and reshape them to maintain consistent channel dimensions. The reshaped features are then concatenated and fed into the reconfiguration transformer block to establish long-range dependencies and local contextual correlations.

After passing through $d$ reconfiguration transformer blocks, the tokens are divided and restored to their original shapes at each stage before being sent to a transformer-based
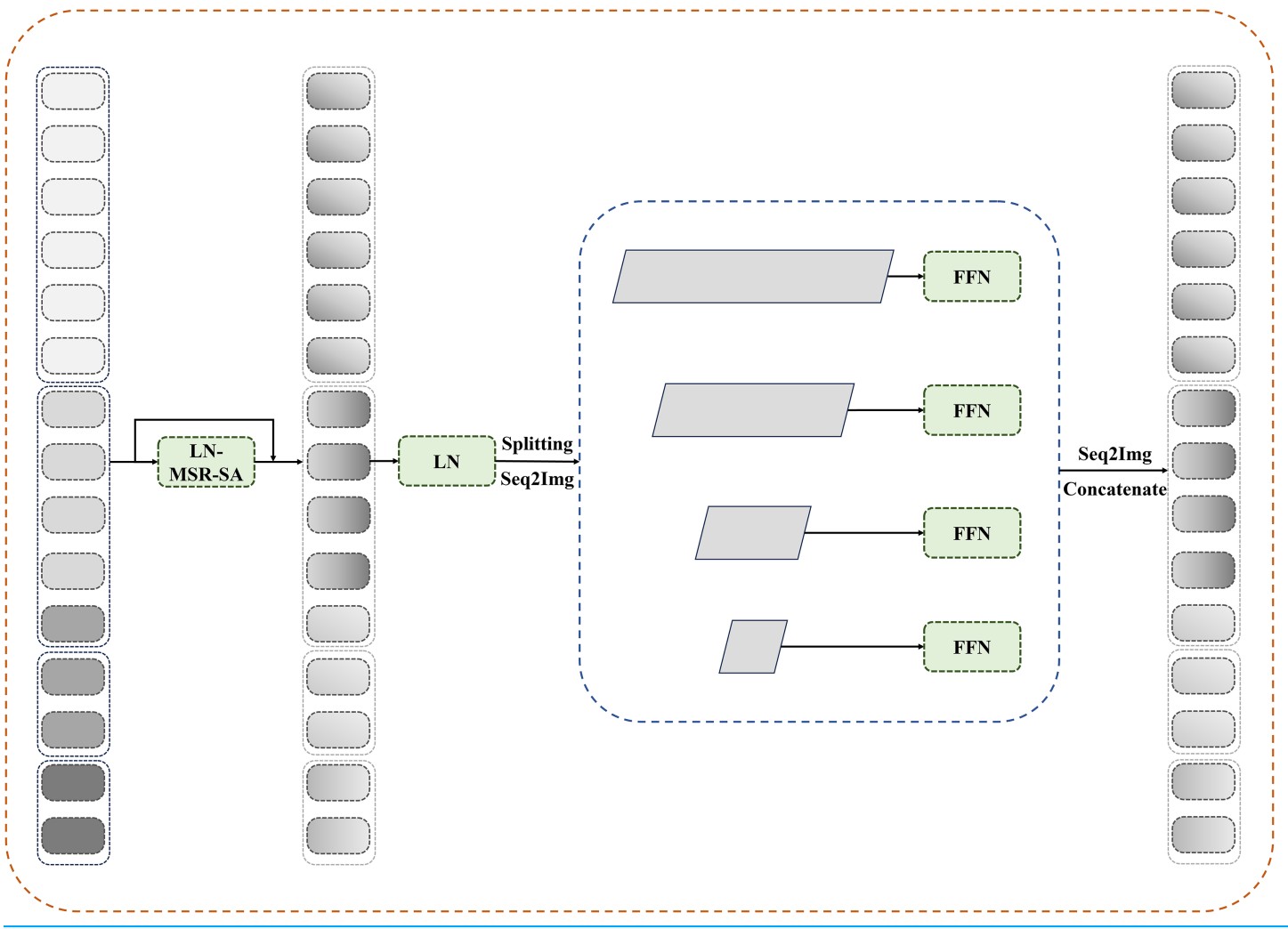

**Figure 3** Details of reconfiguration transformer context bridge block.

decoder. Pixelwise segmentation maps are then predicted by incorporating the corresponding skip connections from the encoder. In this study, the number of loops in the bridge module is set to four.

## Loss function

During the training process for the Synapse dataset, we employed a hybrid loss function that combined the dice loss and cross-entropy loss to address the issue of class imbalance. For the ACDC dataset, we used only the dice loss to optimize our model. The dice loss ($L_{dice}$), cross-entropy loss ($L_{ce}$), and hybrid loss ($L$) are defined as follows:

$$L_{dice} = 1 - \sum_{k=1}^{K} \frac{2w_k \sum_{i}^{N} p(k,i)g(k,i)}{\sum_{i}^{N} p^2(k,i) + \sum_{i}^{N} g^2(k,i)}, \tag{12}$$

$$L_{ce} = -\frac{1}{N} \sum_{i=1}^{N} g(k,i) * log(p(k,i)) + (1 - g(k,i)) * log(1 - p(k,i)), \tag{13}$$

$$L = \lambda L_{dice} + (1 - \lambda)L_{ce}, \tag{14}$$

where $N$ is the number of pixels, $g(k,i)$ and $p(k,i)$ represent the ground truth label and the predicted probability for category $k$, respectively. $K$ is the number of classes, and $\sum_k w_k = 1$. $\lambda$ is a weighting factor that balances the contributions of dice and cross-entropy losses. In this study, based on the work of _Huang et al. (2023)_, $w_k$ and $\lambda$ are set to $1/K$ and 0.6, respectively.

# EXPERIMENTAL RESULTS

## Implementation details

The proposed MSR-UNet was implemented using PyTorch and trained on two Nvidia GeForce RTX 4,090 GPUs, each with 48 GB of graphics memory. MSR-UNet was pretrained on ImageNet, and moderate data augmentation was applied during the experiments. For the training process, the input image size was set to $224 \times 224$, with an initial learning rate of 0.05, and a poly learning rate strategy was used. The maximum number of training epochs was 400, with a batch size of 24. The optimizer for MSR-UNet was stochastic gradient descent (SGD) with a momentum of 0.9 and a weight decay of $10^{-4}$. More details about parameter setting refer to _Huang et al. (2023)_.

## Datasets applied in this study

In this study, two commonly used datasets from the medical segmentation field, the Synapse multi-organ segmentation dataset (Synapse) and the ACDC dataset, were utilized for testing.

Synapse: The Synapse dataset contains 30 abdominal clinical computed tomography (CT) images, comprising a total of 3,779 axial slices. Among these, 18 samples were allocated to the training set, while the remaining 12 samples were used as the testing set (_Chen et al., 2021_; _Fu et al., 2020_). This dataset targets the segmentation of eight abdominal organs: the abdominal aorta, gallbladder, spleen, left kidney, right kidney, liver, pancreas, and stomach.

ACDC: The ACDC dataset consists of MR images with labels for the right ventricle (RV), left ventricle (LV), and myocardium (MYO) from various patients. The dataset includes 70 training samples, 10 validation samples, and 20 testing samples.

## Evaluation criteria

In this study, the average dice similarity coefficient (DSC) and average Hausdorff distance (HD) were used as the segmentation evaluation criteria. DSC denotes segmentation

accuracy, with higher values indicating better performance. HD denotes edge prediction accuracy, where lower values represent better results.

## Segmentation results

### Segmentation results on synapse dataset

We first display the DSC(↑) *vs.* HD(↓) performance on Synapse dataset in Fig. 4. As can be seen, MSR-UNet effectively balances segmentation accuracy and edge prediction, and achieves best performance compared to the U-Net-like methods.

Comparisons of the segmentation results between the proposed MSR-UNet and other methods conducted on the Synapse multi-organ CT dataset are shown in Table 1 (note that the results of the contrastive methods are obtained directly from the original articles except for MISSFormer, and the best results are highlighted in bold). The results show that the proposed method achieved segmentation accuracies of 79.63% (DSC ↑) and 16.21 mm (HD ↓). The DSC values for the eight organs are also reported. Although our method does not exhibit the best performance in the average DSC value (we achieve the second best performance next to MISSFormer), our method achieves the best in the average HD value among all the contrastive methods. This indicates that our method can produce more accurate edge predictions, likely due to the MSR-SA module's ability to merge multi-scale edge information. While our average segmentation accuracy in DSC is lower than that of MISSFormer, our method outperforms MISSFormer in segmenting the gallbladder, liver, and spleen. This suggests that the MSR-SA module effectively leverages local information for these organs. Note that the results for MISSFormer were reproduced using the default parameters provided by the authors through our own implementation.

Figure 5 shows the segmentation results of various methods on the Synapse dataset. The figure indicates that transformer-based methods often encounter difficulties in clearly delineating segmentation regions, which may be due to the transformer's insensitivity to small-scale objects. In contrast, by incorporating the MSR-SA attention mechanism along with a context-bridging module, the proposed MSR-UNet better captures global and long-range interactive semantic information, leading to more accurate segmentation results.

### Segmentation results on ACDC dataset

Following previous work, such as TransUNet (*Chen et al., 2021*), only average DSC value was used to assess the segmentation performance for ACDC dataset. Comparisons of the segmentation results between MSR-UNet and other methods are shown in Table 2 (also the results of the contrastive methods are obtained directly from the original articles except for MISSFormer, and the best results are highlighted in bold) and Fig. 6. Using MR images as input, the proposed MSR-UNet achieves strong segmentation performance, with an accuracy of 90.33% in terms of the DSC metric, which is 0.18% higher than that of MISSFormer. The results for MISSFormer were reproduced using the default parameters through our implementation. Although the overall performance improvement may seem marginal, our method demonstrates higher segmentation accuracy for the RV and LV compared to MISSFormer. This indicates that the proposed MSR-UNet exhibits superior generalizability and robustness across different image types.

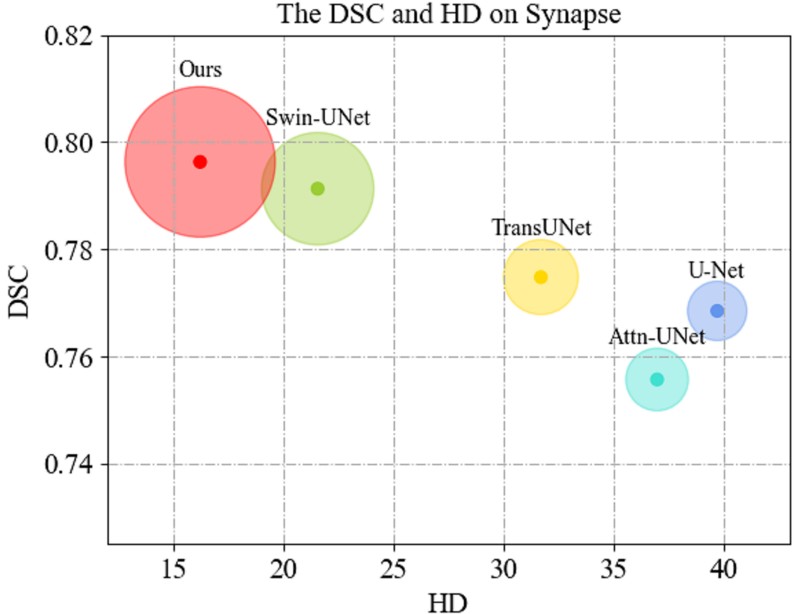

**Figure 4 Comparison of segmentation results of the proposed MSR-UNet and other methods.**

**Table 1 Segmentation comparisons of various methods on Synapse dataset.** The best results are highlighted in bold.

| Methods | DSC↑ (%) | HD↓ (mm) | Aorta | Gallbladder | Kidney (L) | Kidney (R) | Liver | Pancreas | Spleen | Stomach |
|---|---|---|---|---|---|---|---|---|---|---|
| U-Net (*Ronneberger, Fischer & Brox, 2015*) | 76.85 | 39.70 | 89.07 | 69.72 | 77.77 | 68.60 | 93.43 | 53.98 | 86.67 | 75.58 |
| R50 ViT (*Ronneberger, Fischer & Brox, 2015*) | 71.29 | 32.87 | 73.73 | 55.13 | 75.80 | 72.20 | 91.51 | 45.99 | 81.99 | 73.98 |
| Attn-UNet (*Oktay et al., 2018*) | 77.77 | 36.02 | **89.55** | 68.88 | 77.98 | 71.11 | 93.57 | 58.04 | 87.30 | 75.75 |
| TransUNet (*Chen et al., 2021*) | 77.48 | 31.69 | 87.23 | 63.13 | 81.87 | 77.02 | 94.08 | 55.86 | 85.08 | 75.62 |
| Swin-UNet (*Cao et al., 2022*) | 79.13 | 21.55 | 85.47 | 66.53 | 83.28 | 79.61 | 94.29 | 56.58 | 90.66 | **79.60** |
| MISSFormer (*Huang et al., 2023*) | **80.69** | 19.71 | 81.78 | 70.12 | **85.23** | 80.45 | 94.23 | **64.45** | 89.96 | 79.36 |
| DARR (*Fu et al., 2020*) | 69.77 | – | 74.74 | 53.77 | 72.31 | 73.24 | 94.08 | 54.18 | 89.90 | 45.96 |
| V-Net (*Milletari, Navab & Ahmadi, 2016*) | 68.80 | – | 75.34 | 51.87 | 77.10 | **80.75** | 87.84 | 40.05 | 80.56 | 56.98 |
| Ours | 79.63 | **16.21** | 85.55 | **71.44** | 82.63 | 77.86 | **96.76** | 51.25 | **92.45** | 79.14 |

## Ablation study

Ablation studies were conducted on the Synapse dataset to investigate the impact of various factors on the performance of the model. Specifically, we discussed the effectiveness of the MSR-SA module, the impact of the dilation rate, and the contribution of the cross-window fusion operation.

### Effectiveness of MSR-SA

Our framework is based on MISSFormer (*Huang et al., 2023*). To demonstrate differentiation, we replaced the attention mechanism in MISSFormer with ordinary ViT

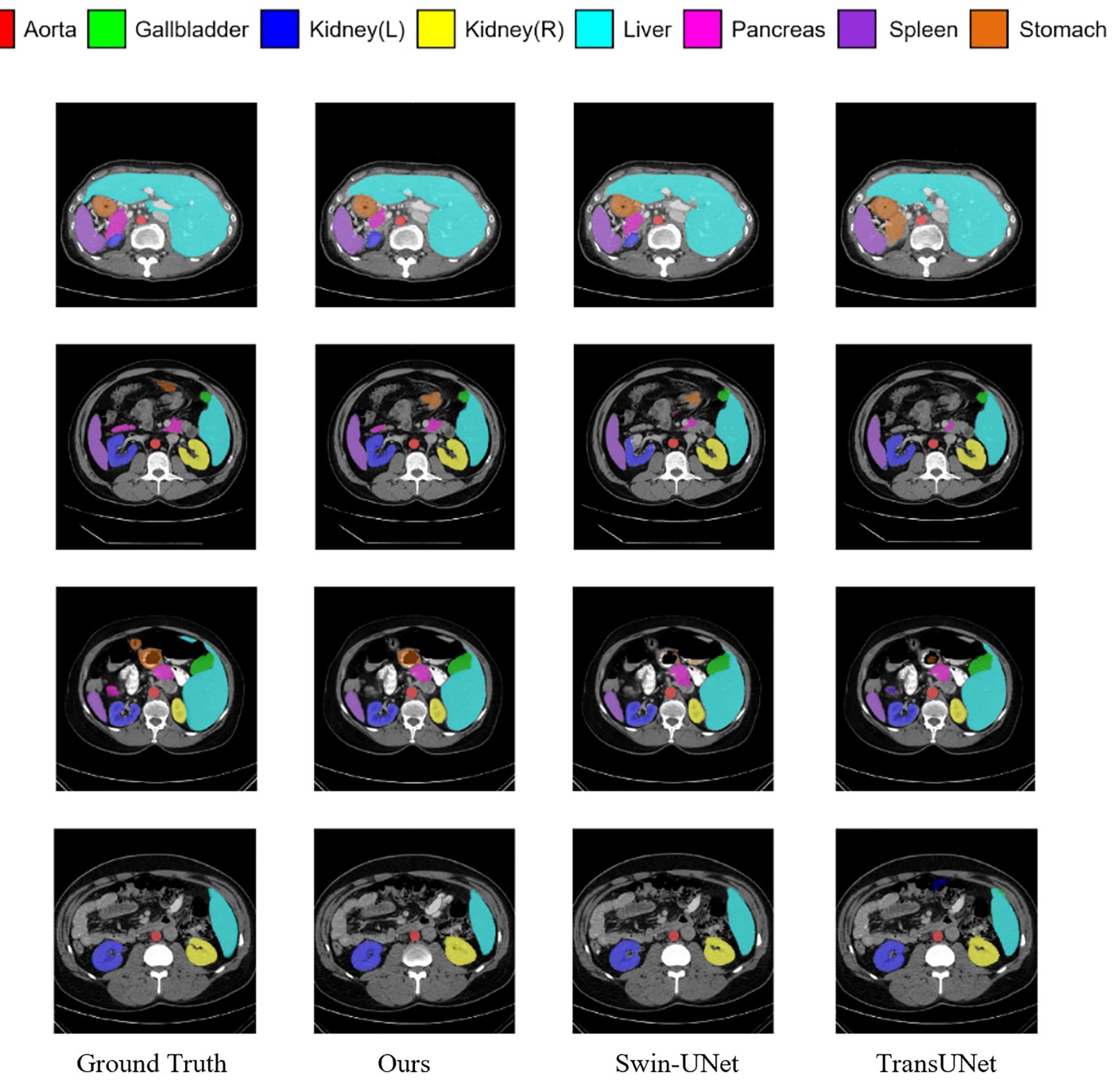

**Figure 5 Visualization of segmentation results of different methods on Synapse dataset.**

(*Dosovitskiy et al., 2020*), Swin-Transformer (*Liu et al., 2021*), and our MSR-SA, while keeping the remaining network configuration unchanged. As shown in Table 3, although MSR-SA did not achieve the same performance as the original MISSFormer in terms of the DSC metric, it demonstrated significant superiority in the HD metric, with an improvement of 3.5%. In addition, the DSC values of our method were 0.57% and 1.81%

**Table 2 Segmentation comparisons of various methods on ACDC dataset.** The best results are highlighted in bold.

| Methods | DSC↑ (%) | RV | MYO | LV |
|---|---|---|---|---|
| R50 U-Net (*Ronneberger, Fischer & Brox, 2015*) | 87.55 | 87.10 | 80.63 | 94.92 |
| R50 Att-UNet (*Ronneberger, Fischer & Brox, 2015*) | 86.75 | 87.58 | 79.20 | 93.47 |
| R50 ViT (*Ronneberger, Fischer & Brox, 2015*) | 87.57 | 86.07 | 81.88 | 94.75 |
| TransUNet (*Chen et al., 2021*) | 89.71 | 88.86 | 84.53 | 95.73 |
| Swin-UNet (*Cao et al., 2022*) | 90.00 | 88.55 | 85.52 | 95.83 |
| MISSFormer (*Huang et al., 2023*) | 90.15 | 88.66 | **87.28** | 94.51 |
| Ours | **90.33** | **88.74** | 85.97 | **96.28** |

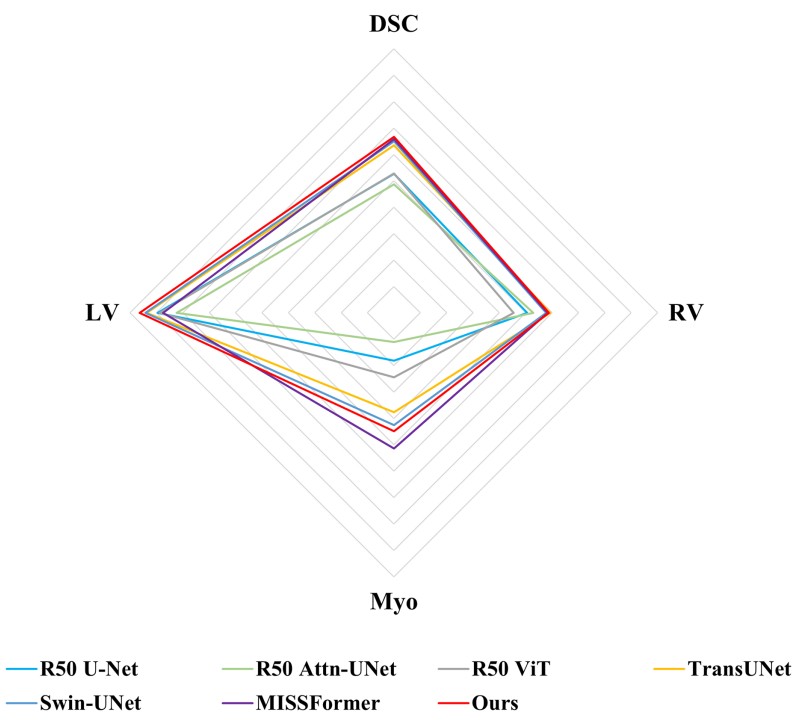

**Figure 6 Radar chart for attribute-based assessment of DSC values on ACDC dataset.**

**Table 3 Segmentation comparisons of different attention mechanisms with same backbone conducted on Synapse dataset.**

| Methods | DSC↑ (%) | HD↓ (mm) |
|---|---|---|
| MISSFormer (Original) | 80.69 | 19.71 |
| MISSFormer-swin-transformer | 79.06 | 19.34 |
| MISSFormer-ViT | 77.82 | 28.65 |
| MISSFormer-MSR-SA (Ours) | 79.63 | 16.21 |

**Table 4 Segmentation results with different dilation rates conducted on the Synapse dataset.**

| Methods | DSC↑ (%) | HD↓ (%) |
|---|---|---|
| MSR-SA-low | 79.29 | 17.34 |
| MSR-SA-high | 78.62 | 16.65 |
| MSR-SA | 79.63 | 16.21 |

**Table 5 Segmentation results of different cross-window fusion methods conducted on the Synapse dataset.**

| Methods | DSC↑ (%) | HD↓ (mm) |
|---|---|---|
| Linear | 77.12 | 22.36 |
| Average pooling | 79.56 | 19.47 |
| Max pooling | 79.63 | 16.21 |

higher than those achieved by the Swin-transformer and ViT, respectively. These results indicate that our MSR-SA outperforms other commonly used self-attention methods in terms of segmentation accuracy and edge precision.

### Impact of dilation rate

In MSR-SA, we prioritize aggregating the multi-scale information embedded in the grouped features. To further evaluate the impact of multi-scale aggregation, we investigated two additional models: MSR-SA-low and MSR-SA-high. MSR-SA-low sets the dilation rate of each group to 1, resulting in the extracted query containing only single-scale information. In contrast, MSR-SA-high assigns a local window size equivalent to the feature map, capturing more scale information. As shown in Table 4, the proposed MSR-SA consistently outperformed both MSR-SA-low and MSR-SA-high, underscoring the effectiveness of aggregating multi-scale information in MSR-SA.

### Cross-window fusion

In the cross-window fusion step, MSR-SA adopts a symmetric aggregation function $\sigma(\cdot)$ to merge the features of the sampling points at the same positions across different windows. In practical applications, average pooling is used as the default fusion operator. To identify the optimal fusion method, three operators—linear, average pooling, and maximum pooling—were tested using MSR-UNet as a benchmark. The segmentation accuracies of these three fusion operators were assessed on the Synapse dataset. The comparison results, presented in Table 5, indicate that maximum pooling achieved the best performance. Consequently, maximum pooling was selected for the final implementation.

## CONCLUSIONS

This study proposed a multi-scale reconfiguration self-attention (MSR-SA) module to investigate the optimal tradeoff between multi-scale information and long-range dependency in a medical image segmentation model. By decomposing attention matrix

into a series of sparse sub-matrix, MSR-SA integrates features of different granularities. The attention heads are divided into multiple groups, and each group is assigned an increased dilation rate and partitioned into local windows. Then a reconfiguration operation was introduced to simultaneously capture both long-range and multi-scale information by gathering keys through dilated sampling. With the integration of MSR-SA, long-range dependencies were exploited with a low computational burden on the local windows. Benefiting from the modeling capabilities of multi-scale and long-range correlations provided by the MSR-SA module, the proposed multi-scale reconfiguration U-Net (MSR-UNet) demonstrated strong performance in complex medical image segmentation tasks. Future work will focus on simplifying the attention module and optimizing parameters to achieve a more efficient model design. To explore channel features from variant scales, multi-scale channel attention module needs to be further investigated. Moreover, the state space model such as Mamba can be used to reduce the quadratic complexity of transformer-based methods.

## ABBREVIATIONS

| | |
|---|---|
| **MSR** | Multi-Scale Reconfiguration |
| **SA** | Self-Attention |
| **CT** | Computed Tomography |
| **LV** | Left Ventricle |
| **DSC** | Dice Similarity Coefficieng |
| **ACDC** | Automated Cardiac Diagnosis Challenge dataset |
| **SGD** | Stochastic Gradient Descent |
| **RV** | Right Ventricle |
| **MYO** | MYOcardium |
| **HD** | Hausdorff Distance |

## ACKNOWLEDGEMENTS

The authors would like to thank Editage for English language editing. We would also like to thank the anonymous reviewers for their valuable comments.

### Funding

This work was supported by the Natural Science Foundation of Hebei Province (No. F2022201013); the Science and Technology Program of Huaibei (No. 2023HK037); the Anhui Shenhua Meat Products Co., Ltd. Cooperation Project (No. 22100084); the Anhui Provincial College Student Innovation and Entrepreneurship Training Program (No. S202410373029); and the Entrusted Project by Huaibei Mining Group (2023). There was no additional external funding received for this study. The funders had no role in study design, data collection and analysis, decision to publish, or preparation of the manuscript.

## Grant Disclosures

The following grant information was disclosed by the authors:

Natural Science Foundation of Hebei Province: F2022201013.

Science and Technology Program of Huaibei: 2023HK037.

Anhui Shenhua Meat Products Co., Ltd. Cooperation Project: 22100084.

Anhui Provincial College Student Innovation and Entrepreneurship Training Program: S202410373029.

Entrusted Project by Huaibei Mining Group (2023).

## Competing Interests

The authors declare that they have no competing interests.

## Author Contributions

- Shuai Wang conceived and designed the experiments, performed the experiments, performed the computation work, prepared figures and/or tables, and approved the final draft.
- Lei Liu conceived and designed the experiments, performed the computation work, authored or reviewed drafts of the article, and approved the final draft.
- Jun Wang analyzed the data, authored or reviewed drafts of the article, and approved the final draft.
- Xinyue Peng performed the experiments, performed the computation work, prepared figures and/or tables, and approved the final draft.
- Baosen Liu conceived and designed the experiments, analyzed the data, prepared figures and/or tables, and approved the final draft.

## Data Availability

The code is available at GitHub and Zenodo:

- https://github.com/davidsmithwj/MSR-UNet

- JUN WANG. (2024). davidsmithwj/MSR-UNet: v1.0.0 (v1.0.0). Zenodo. https://doi.org/10.5281/zenodo.13969855.

The Synapse dataset is available at: https://doi.org/10.7303/syn3193805.

The ACDC dataset is available at https://humanheart-project.creatis.insa-lyon.fr/database/#collection/637218c173e9f0047faa00fb.

## Supplemental Information

Supplemental information for this article can be found online at http://dx.doi.org/10.7717/peerj-cs.2563#supplemental-information.

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
