# Peer review of "MSR-UNet: enhancing multi-scale and long-range dependencies in medical image segmentation"

_PeerJ Computer Science, doi:10.7717/peerj-cs.2563_

## Round 0.1 · original submission · Major Revisions

· Academic Editor

Major Revisions

I am happy to announce that the review of your manuscript is now complete. Kindly revise the manuscript as per the reviewer suggestions and resubmit it.

Reviewer 1 ·

Basic reporting

+ Reference must be improved. Most of references should be cited from recent researches (not over five years from present)
+ In Conclusion, there should have several future works so that readers could follow the potential developments
+ It seems to be strange that Fig. 1 appears in the Introduction section. Maybe, it should appear in section 2 or last section.
+ When you compare your method with the others, you might give a comment if those studies were repeated

Experimental design

+ There are many abbreviations in this section. To support readers, one Table for summarization of these symbols should be produced for readers to easily understand
+ In Table to display the quantities of research, unit of each parameter should be announced

Validity of the findings

+ Since authors compare your works, the name of method which presented in Table, should be displayed with the cited reference
+ Authors should explain briefly why you want to compare the segmentation result or ablation (rate and fusion)

Reviewer 2 ·

Basic reporting

- In this manuscript, authors present "MSR-SA" for enhancing multi-scale and long-range dependencies in medical image segmentation.

- This manuscript is easy to follow, but needs improvement.

- Add a paragraph in the end of Introduction which highlights the remaining portion of this manuscripts.

- Add a sub-section called "Our Contributions" in the Introduction section.

- Highlight current status and limitation of current literatures in the end of Related works.

- Also, highlight the novelty of this study.

Experimental design

- Describe datasets information in a table.

- Add a sub-section to describe evaluation metrics.

- Highlights short details about each comparable methods.

- Add a discussion section in the experimental section to give more insights about results.

- Include error analysis found in results as a sub-section.

Validity of the findings

- Novelty should be highlighted with comparable methods which explain the current parameters setting of MSR-SA are performing good as compared to other methods.

- Conclusion should be in more detail.

- Add a sub-section which refers to future directions.

Additional comments

Needs major revision before acceptance.

Reviewer 3 ·

Basic reporting

All comments have been added in detail to the last section.

Experimental design

All comments have been added in detail to the last section.

Validity of the findings

All comments have been added in detail to the last section.

Additional comments

Review Report for PeerJ Computer Science
(MSR-UNet: Enhancing multi-scale and long-range dependencies in medical image segmentation)

1. Within the scope of the study, a deep learning model called MSR-UNet based on Unet has been proposed for medical image segmentation.

2. In the introduction section, medical image segmentation, the effect of vision transformers and the importance of the study are clearly mentioned in items.

3. The Related works section is addressed from three different perspectives as Multi-scale Feature Integration, Attention Mechanism and Medical Image Segmentation, and this section definitely needs to be detailed. It is suggested to add a literature table consisting of columns such as "used dataset, originality, pros and cons, metrics, results" to this section.

4. When the proposed MSR-UNet framework and its content and the proposed MSR-SA module are examined in detail, it is observed that it has a certain level of originality. In this section, it should be detailed why the loss function used was preferred, what its differences are from other loss functions used in similar studies in the literature and what its pros are.

5. It is observed that the Implementation Details are explained at the necessary level. However, since the determination of hyperparameters can have a positive/negative effect on the result, it is important to explain this part. In this context, how the parameters such as learning rate, epoch, batch size preferred for the study were determined and whether different experiments were performed should be explained in more detail.

6. When the results obtained and the metrics used are examined, it is observed that they are at a sufficient level.

As a result, although the study has the potential to make a significant contribution to the literature, the sections mentioned above should definitely be taken into consideration.

---

## Round 0.2 · accepted · Accept

· Academic Editor

Accept

The author has addressed the reviewer's comments properly. Thus I recommend publication of the manuscript.

Reviewer 2 ·

Basic reporting

- Authors have addressed comments in the revised manuscript.

Experimental design

Authors have addressed comments related to experiments

Validity of the findings

Authors have addressed all comments

Additional comments

Authors have addressed all comments

Reviewer 3 ·

Basic reporting

All comments have been added in detail to the last section.

Experimental design

All comments have been added in detail to the last section.

Validity of the findings

All comments have been added in detail to the last section.

Additional comments

Review Report for PeerJ Computer Science
(MSR-UNet: Enhancing multi-scale and long-range dependencies in medical image segmentation)

Thanks for the revision. The new changes in the paper and the responses to the reviewer comments are sufficient. I recommend that the paper be accepted due to its potential to make a significant contribution to the literature and its final form. I wish the authors success in their future papers and projects.